# Internet, Green Innovation and Industrial Upgrading

Lei Tong and Yafei Rong *

School of Management, China University of Mining and Technology (Beijing), Beijing 100083, China
* Correspondence: tsp1600501022@student.cumtb.edu.cn

**Abstract:** The internet and green innovation are important driving forces to promote industrial high-quality and sustainable development at present. Studying their independent and interactive effects on industrial upgrading is of great practical and theoretical significance. Based on the panel data of 30 provinces in China from 2006 to 2019, this paper constructs a spatial error model based on four weight matrices to study independent and interactive effects of the internet and green innovation on industrial upgrading. The results show that the internet, green innovation, and industrial upgrading have significant spatial correlation, with all showing high–high and low–high agglomeration trends. Both the internet and green innovation contribute to industrial upgrading, and their interaction effect is more significant for enhancing industrial upgrading. The heterogeneity analysis finds that green innovation has a greater impact on industrial upgrading in eastern China, and the internet in the central and western regions has a greater impact on industrial upgrading. Few previous studies put the internet, green innovation, and industrial upgrading into the unified framework. This paper expands and enriches the research on the relationship among the three to analyze the independent and interactive effects of the internet and green innovation on industrial upgrading by spatial metrology. We also promote the construction, promotion, and application of the internet and optimization of the green innovation environment, taking the "Internet plus green innovation" strategy as the foothold and implementing differentiation and a dynamic strategy that provides a reference for how to realize the transformation and upgrading of China's industrial structure through the internet and green-innovation strategy.

**Keywords:** internet; green innovation; industrial upgrading; interaction effect; spatial metrology

## 1. Introduction

Green innovation as an innovation from the environmental perspective has become a new and important way to break the conflict between economic growth and environmental pollution since the Fifth Plenary Session of the 18th Communist Party of China Central Committee put forward "innovation", "coordination", "green", "open", and "shared" as five development concepts [1]. Therefore, how to fully utilize green innovation to achieve high-quality and sustainable development of industries in the current stage of a critical period of economic and industrial transformation [2] is an important question. At the present time, there are many challenges to green innovation to be faced for enterprises in China. Firstly, green innovation has a lower success rate and higher risk, and requires more capital investment. Secondly, China's market and institution on green innovation is imperfect. This might lead to green innovation not being able to drive industrial upgrading or having less promotional effect in reality. Given that the involvement of the internet can make the market more transparent, reduce costs, and improve the success rate of green innovation [3], the following questions are raised: Can green innovation become the key choice for industrial upgrading? What is the mechanism of green innovation and industrial upgrading? What role does the internet play between green innovation and industrial upgrading? What is the mechanism of the internet and green innovation on industrial upgrading? In order to answer the above questions, this paper uses the spatial econometric model to measure both independent effect and interactive effect of the internet and green

innovation on industrial upgrading using a sample of 30 provinces in China from 2006 to 2019.

The remainder of the paper is organized as follows. Section 2 includes a literature review. Section 3 presents theoretical mechanisms and theoretical hypotheses. Section 4 presents the model establishment and variable measurement. Section 5 reports the empirical results and corresponding analysis. Section 6 presents the conclusion and discussion.

## 2. Literature Review

The research literature related to the internet, green innovation, and industrial upgrading is primarily divided into the following two categories:

The first category encompasses research on the concept definition and measurement of the internet, green innovation, and industrial upgrading. As for the measurement of internet indicators, scholars have different understandings, mainly including the single-index method and the composite-index method. Among them, the single-index method refers to one index to measure the internet development, whereas the composite-index method aims to comprehensively understand the development of the internet from multiple perspectives. Due scholars usually studying the internet from different perspectives and based on different purposes, the index systems constructed have great differences. Although the single metric is often criticized, it is the most commonly used measure [4]. Secondly, green innovation is an effective combination point of innovation and green development and, based on the premise of low carbon, energy conservation, and ecological protection, aims to reduce the environmental impact of any new elements in the organization compared with traditional innovation. Although different scholars have different concepts of green innovation, they all affirm the role of green innovation in environmental improvement [5]. Green innovation can be defined as innovations of products, processes, or management practices aimed at reducing environmental impacts [6]. Most scholars use stochastic frontier analysis (SFA) and data envelopment analysis (DEA) to evaluate green-innovation efficiency [5,7]. Among them, due SFA needing to assume the estimation model in advance and not being able to handle both good and bad outputs at the same time, more scholars tend to choose DEA to measure green-innovation efficiency.

For industrial upgrading, the concept and measurement vary considerably among scholars at present because industrial upgrading is a dynamic process and the intension and extension is continuing to develop and enrich with social progress [8]. One of the viewpoints that scholars generally agree on is that industrial upgrading can be divided into industrial advancement and industrial rationalization [9]. Among them, industrial advancement is a process switching from a low to high development level and industrial rationalization is a process of inter-industrial continuous coordination and optimization. Most existing studies on industrial upgrading focus on whether industrial-structure adjustment brings a structural bonus [10]. The existing industrial-advancement index mainly reflects the industrial-servitization trend, and the industrial-rationalization index reflects the inter-industrial-coupling coordination [11]. Although this viewpoint provides basic and rigorous macro-analytical thinking, the limitations are also obvious [12]. Industrial structure is not only the converter of natural resource inputs but also the control body of the quality and type of pollutants [13]. Thus, Yang et al. redefined the industrial-upgrading intension and extension based on a high-quality and sustainable perspective [14]. Industrial rationalization is redefined as the aggregation quality of inter-industrial organic correlation, reflecting the inter-industrial coordination degree and effective utilization of resources. Industrial advancement is mainly reflected in the orientation of the economy structure [15]. Driven by the IT revolution, the orientation of the economy structure for industrial upgrading includes two trends, industrial servitization and industrial intelligence [16]. Therefore, based on the study by Yang et al., industrial upgrading can be divided into industrial advancement and industrial rationalization, in which industrial advancement has two aspects, industrial servitization and industrial intelligence. Industrial rationalization in-

cludes inter-industrial coordination and resource utilization (in this paper, we call resource utilization "industrial greening").

The second category encompasses research on the relationship among internet, green innovation, and industrial upgrading. In terms of research on the relationship between internet and industrial upgrading, three different views have been formed. First, scholars generally believe that the internet can significantly promote industrial upgrading. For instance, Lu et al. took Shanghai as a research object and found that the internet is the Granger cause to promote industrial upgrading [17]. Huang et al. also took Shanghai as a research object and pointed out that the internet can promote industrial advancement, rationalization, and greening development as a clear resource [18]. The research of Liu et al. showed a similar viewpoint based on the panel data of 31 provinces in China [19]. Shi et al. thought that the penetration of the internet can significantly promote the labor division of manufacturing enterprises in China by reducing the search cost [20]. Lu et al. [21] also found that the internet promotes industrial upgrading by reducing the search cost based on the theoretical models of the internet, transaction costs, and productive services. Second, a few studies discovered that the internet and industrial upgrading are mutually causal [22]. Studies by Huang et al. also agreed with this view, but the empirical analysis did not pass the test [18]. Cui et al. [23] studied the interaction between internet infrastructure and industrial upgrading and found empirical results like those in the study by Huang et al. The third is that the internet might cause some harmful effects on industrial upgrading. Some problems such as "production paradox", "Baumol disease", "wage polarization", might result from the internet for industrial upgrading [24]. The study by Xu et al. [25] proved this viewpoint and found that the internet can only promote industrial advancement and not be conducive to rationalization. The reason is that the different integration degree of internet technology in three major industries is owed to improper convergence of productivity in three major industries. Thus, there is still a question regarding the relationship between the internet and industrial upgrading in China.

In terms of research on the relationship between green innovation and industrial upgrading, most studies focus on the influence of green innovation on industrial upgrading. For example, Wang et al. used a panel fixed-effect model to study green innovation and industrial upgrading in the Yangtze Delta Region and found that green innovation could significantly promoted industrial upgrading [7]. Studies by Gao et al. and Xie et al. illustrated similar conclusions based on the panel data of 31 provinces in China [26,27]. Ge et al. [2] studied the effect of renewable-energy technology innovation, one type of green innovation, on industrial upgrading and found that it contributed significantly to the adjustment of industrial structure. He [28] found that green innovation played a role in improving product cleanliness and enterprise production efficiency, which could be the endogenous power for the industry to achieve green sustainable-growth transformation. Besides, some scholars indirectly agreed on the promotion of green innovation on industrial upgrading while studying other issues. For example, Guo et al. [29] pointed out that green innovation is an important transmission mechanism for the digital economy to release the dividend of industrial-structure upgrading. Gao et al. found that industrial upgrading took a mediating effect when it came to green innovation and the carbon-emissions nexus in China [30]. They all agreed that green innovation is a kind of motivation for industrial upgrading. Different from the above research, studies by Qian et al. and Zhang et al. found that green innovation and industrial upgrading were mutually causal and estimated coupling coordination [1,31]. Due the mutual causality not being confirmed with an empirical test, Liu et al. explored the relationship between them based on a dynamic simultaneous equation [32]. The empirical result showed that green innovation only unidirectionally and significantly promoted industrial upgrading. Based on the above research, we gained some theoretical and empirical support for our study and found that a study on the spatial spillover of green innovation on industrial upgrading might further enrich existing research.

In terms of the relationship between the internet and green innovation, most scholars only focus on the unilateral impact of the internet on green innovation. For instance, Li et al. studied how the development of the internet promotes green innovation from the perspective of patents [33]. They illustrated that the internet mainly contributes to green innovation by reducing transaction costs and improving technology research and development. Han et al. supplemented that by putting forth that the internet may also make the polluting behavior of enterprises more transparent, forcing enterprises to adopt green-innovation technology to solve the pollution problem from the source [34]. Wang et al. [35] regarded that development of the internet could significantly promote green innovation in China directly and indirectly by means of promoting producer-services agglomeration, driving financial development and reducing resource dependence. Chen et al. [5] agreed with the viewpoint of Liu et al. and illustrated that the internet promotes green-innovation efficiency by reducing transaction costs. They found that the efficiency of green innovation has significant characteristics of spatial spillover. Studies by Hu et al. and Fang et al. analyzed the spatial spillover of the internet on green innovation [36,37] and concluded with a similar result: The internet has spillover effects on green innovation. Among them, Fang et al. explained the mechanism of the internet on green innovation and thought that the internet can promote the flow and cross-border integration of elements, especially R&D innovation elements, to help to reduce information asymmetry and transaction costs. Tang et al. [38] reported a similar result, that broadband infrastructure accelerates the knowledge and technology spillover regarding environmental protection, reducing the flow cost of green-innovation elements and thus helping to promote green-technology innovation.

In fact, the internet as an innovative technology can interact with green innovation. The internet is also popularized and developed while promoting the development of green innovation. For green innovation, the internet can be used as an information carrier of innovation subjects, which not only provides rich learning resources but also broadens the information-exchange platform aside from the traditional way [34]. For the internet, green innovation is a main motivation that can not only provide technical support for the development and upgrading of the internet but also improve the bottleneck problem of underlying technology and processes [39]. However, most studies only indirectly refer to the interaction between the internet and green innovation when discussing relevant issues. For example, Xu et al. [40] confirmed the interaction relationship between the internet and green innovation when discussing the green industrial internet. The existing institutional environment could force enterprises to adopt green innovation. The green industrial internet as an innovation technology of green innovation and internet development can decarbonize and optimize environmental protection and monitoring and reduce operational costs and energy usage for long-term growth. China formulated the "Internet plus" action in 2015 [36]. The relevant policies on "Internet plus green innovation" have been issued by governments in China. For instance, the Ministry of Science and Technology of China increased the "Internet+"-related information transmission and internet development-level index in the China Regional Innovation Capability Monitoring Report 2015 index system. In February 2016, the National Development and Reform Commission issued the Guiding Opinions on Promoting the Development of "Internet+" Smart Energy to promote the new models and new business forms of the energy internet. The General Office of the State Council issued the Opinions on Deeply Implementing the Action Plan of "Internet + Circulation" in 2016. The Opinions clearly put forward a goal to develop green circulation and consumption and promote green goods. Besides, under the internet ecological environment, some enterprises have also begun to explore how to realize product innovation by relying on the internet ecology. For instance, NIO has established the integration function of internet technology and an internet platform, and MI uses the latest internet tools and platforms to encourage customers to participate in MI R&D. Thus, studying the mechanism of coordination between the internet and green innovation is significant.

Based on the existing literature and expert opinions, we found that the studies mainly focus on the impact of the internet on green innovation and the impact of green innovation on industrial upgrading. The mechanism by which the internet and green innovation interact and affect industrial upgrading is unclear. The spatial spillover of green innovation on industrial upgrading should be investigated, which can enrich related research. Thus, based on the existing research, we have further enriched the theoretical and empirical research on the internet, green innovation, and industrial upgrading. The main contributions are as follows. First, this paper firstly incorporates the internet, green innovation, and industrial upgrading into the same research framework to analyze the theoretical mechanisms of the internet, green innovation, and their interaction on industrial upgrading. Secondly, the spatial pattern of the internet, green innovation, and industrial upgrading based on the super-efficiency slack-based model of unexpected output, coupling-coordination-degree model, and spatial correlation analysis is discussed. Thirdly, we use spatial metrology to measure both the independent effect and the interactive effect of the internet and green innovation on industrial upgrading. In view of the sensitivity of the spatial economic model to the spatial weight matrix, four spatial weight matrices—the adjacency matrix, geographical matrix, economic matrix, and nested matrix—are constructed to estimate the impact of the internet and green innovation on industrial upgrading. Meanwhile they verify the stability of the result. Fourthly, we analyze the regional differences across the three regions of eastern, central, and western China.

## 3. Theoretical Mechanisms and Theoretical Hypotheses

The above research provides a rich analysis perspective and ideas for this paper and is an important basis for further discussion of the mechanism. On this basis, we found that the theoretical mechanism of the internet, green innovation, and industrial upgrading is mainly based on endogenous-growth theory, synergy theory, and sustainable development theory. First of all, the internet and green innovation are both the core and source of achieving sustainable industrial development, which can promote industrial upgrading through knowledge spillovers and human capital, reducing transaction costs. Secondly, the internet and green innovation have an interaction mechanism that can produce a "1 + 1 > 2" effect on industrial upgrading. Therefore, the theoretical mechanism of this paper includes three aspects: the theoretical mechanism of the internet to promote industrial upgrading, the theoretical mechanism of green innovation to promote industrial upgrading, and the theoretical mechanism of their interaction to promote industrial upgrading. The specific impact path is shown in Figure 1.

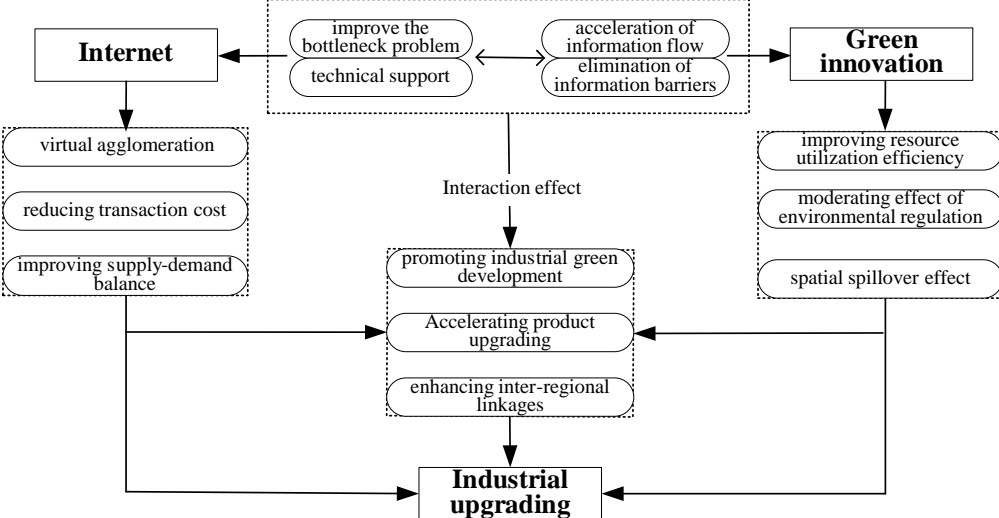

**Figure 1.** The theoretical framework of the core variables.

### 3.1. The Internet Promotes Industrial Upgrading

The internet can promote industrial upgrading through the virtual-agglomeration effect. The virtual realm constructed by the internet can form a virtual agglomeration based on real-time data and information exchange, which is similar to traditional geographic agglomeration, and promotes industrial upgrading through Marshall, Jacob, and Porter externalities [41]. Secondly, transaction costs can be efficiently reduced by the internet for industrial upgrading. On the one hand, enterprises can easily obtain information such as credit status, contract performance, technological level, product quality, and reputation on the internet, which can effectively reduce the searching cost [20]. On the other hand, the information exchange can be more convenient and rapid on the internet, which significantly decreases the coordination cost. Third is that the internet can promote industrial upgrading by improving the balance between supply and demand [42]. On the demand side, the internet firstly improves consumption structure in order to enhance consumption upgrading. Then, the industry-pulling effect and income-elasticity effect of consumption-upgrading affects promote industrial upgrading. On the supply side, enterprises can rely on the internet to mine and analyze consumer-demand information, which can decrease the uncertainty and risk of market need. Thus, the efficiency of resource utilization and scientific allocation are enhanced.

**Hypothesis 1.** *The internet can promote industrial upgrading by virtual agglomeration, reducing transaction cost and improving supply–demand balance.*

### 3.2. Green Innovation Promotes Industrial Upgrading

Green innovation generally affects industrial upgrading through the following three theoretical mechanisms. The first is the improvement of resource-utilization efficiency. To be specific, green innovation can increase the marginal productivity of labor [43]. When green innovation comes to improve traditional industry or develop emerging industry, firms will ask employees to gain knowledge and learn technology to obtain competitiveness. On the other hand, green innovation can integrate and allocate visible resources such as labor and capital, which not only promote the upgrading of high-energy-consumption and -pollution sectors or industries but also guide the flow of resources to sectors or industries with higher production efficiency or technology intensity [44]. The second is the moderating effect of environmental regulation. As environmental regulations gradually become stricter, firms likely face the situation of the cost of pollution-treatment approaches or exceed the cost of technology research and development, which prompts enterprises to utilize green-innovation technology to ensure sustainable development [7,34]. The third is the spatial-spillover effect. The spatial-spillover effect of green innovation can be divided into knowledge- and environmental-spillover effects [45]. The knowledge-spillover effect is mainly the process of knowledge dissemination and diffusion among geographic regions through many commercial or non-commercial ways, which can be acted on in the production process by the imitation effect, communication effect, competition effect, and incentive effect to promote industrial upgrading. The environmental-spillover effect refers to the external effects on the environment generated by green innovation affecting the surrounding areas through demonstration and learning effects [44].

**Hypothesis 2.** *Green innovation can promote industrial upgrading by improving resource-utilization efficiency, moderating the effect of environmental regulation and the spatial-spillover effect.*

### 3.3. The Interaction between Internet and Green Innovation Promotes Industrial Upgrading

The theoretical mechanisms among the internet and green innovation with industrial upgrading can be primarily divided into the following three aspects. The first aspect is promoting industrial green development. The internet can enhance industrial green development by strengthening external supervision and improving environmental-information

governance of innovation subjects. To be specific, one is that the internet can provide a channel for the government and the public to monitor and control environmental information, leading to enterprises being more transparent about their polluting behaviors and the supervision and control ability of government being improved [33,34]. The other is that due to the information having the characteristics of fluidity and strong diffusibility on the internet, open and transparent government information can play a good demonstration-learning effectiveness in order to avoid pollution having an effect due to incomplete enforcement [46]. The second is acceleration of product upgrading. The use of internet technology can accurately identify the market demand for new products and new services and guide the direction of innovation so as to improve the achievement rate, shorten the cycle of innovation achievements into products, and accelerate regional industrial upgrading [47]. The third aspect is the promotion of inter-regional linkages. In the internet economy, element flow across time and space, especially R&D innovation elements, is accelerated, which can further enhance the collaborative innovation effect and regional radiation effect in green innovation [37]. Due the cross-regional linkages compensating for intra-regional supply-side limitations [48], the interaction of the two can enhance local industrial upgrading.

**Hypothesis 3.** *The interaction between the internet and green innovation can promote industrial upgrading by promoting industrial green development, accelerating product upgrading and enhancing inter-regional linkages.*

## 4. Variable Measurement and Model Establishment

### 4.1. Variable Measurement

4.1.1. Explained Variable—Industrial Upgrading

Industrial servitization: Referring to the research of Gan et al. [49], this paper selected the ratio of the added value of tertiary industry and of secondary industry to represent industrial servitization.

Industrial intelligence: In the IT revolution, industrial intelligence is mainly based on digitalization [11,18]. The output value of industrial digitalization (also called the information and communication technology industry) is used to measure industrial intelligence [11].

Inter-industrial coordination: The Theil index was selected to measure the inter-industrial coordination, which can measure the deviation of the output value and employment structure of each industry as well as the difference in economic status of each industry [13], as shown in Equation (4).

Industrial greening: In the current period of accelerated industrialization and urbanization, reducing resource consumption and improving resource-utilization efficiency can effectively resolve the contradiction between population resources and economic development due to China's resource endowment being poor. Because energy consumption is an important cause of environmental pollution, the development and utilization of any kind of energy will have an impact on the environment to varying degrees, and the reduction of energy consumption means the reduction of environmental pollution and the improvement of resource utilization to some extent. Energy-consumption reduction can affect the level of environmental pollution and resource utilization by the necessary clean technologies [50]. The proportion of total energy consumption in GDP is used to measure industrial greening.

Considering that industrial upgrading is a coordinated and unified process, ignoring any aspect means that high-quality and sustainable industrial development cannot be achieved [51]. Based on the practice of Zhang et al. [52], we attempted to use the coupling-coordination-degree model to measure Chinese industrial upgrading, as shown below:

$$upgrade = \sqrt{C \times T}$$
$$= \sqrt{\sqrt[4]{\sqrt[4]{\frac{IS \times II \times IC \times IG}{\left(\frac{IS+II+IC+IG}{4}\right)^4}} \times (0.25IS + 0.25II + 0.25IC + 0.25IG)}} \tag{1}$$



$$C = \sqrt[4]{\frac{IS \times II \times IC \times IG}{\left(\frac{IS+II+IC+IG}{4}\right)^4}} \tag{2}$$

$$T = 0.25IS + 0.25II + 0.25IC + 0.25IG \tag{3}$$

$$IC = \sum_{k=1}^{3} \frac{G_k}{G} \left| \frac{G_k/G}{L_k/L} - 1 \right| \tag{4}$$

where $IS$ represents industrial servitization, $II$ represents industrial intelligence, $IC$ is inter-industrial coordination, $IG$ is industrial greening, $G_k$ represents the output value of the sector $k$, and $L_k$ represents the number of employees in the sector $k$. It should be noted that $IS$, $II$, $IC$, and $IG$ are standardized data by the maximum-difference normalization method. the formulations are shown as follows:

$$IU = \begin{cases} \frac{Indust_{i,t}-\min(Indust_{i,t})}{\max(Indust_{i,t})-\min(Indust_{i,t})}, & IS, II \\ \frac{\max(Indust_{i,t})-Indust_{i,t}}{\max(Indust_{i,t})-\min(Indust_{i,t})}, & IC, IG \end{cases} \tag{5}$$

where $IU$ represents one of $IS$, $II$, $IC$, $IG$, and $Indust$ represents the raw data of a dimension of industrial upgrading.

### 4.1.2. Explanatory Variable—Green Innovation

As a multi-input and multi-output economic process, choosing appropriate input and output indicators of green innovation is crucial to obtaining effective evaluation results. Therefore, the DEA method was used in this paper to evaluate the development of green innovation. Because the non-radial and non-angular super-efficiency SBM model (NNSSBM) can solve the sickness of the influence of slack variables on efficiency and random errors on each subject compared with the traditional radial DEA method, this paper used NNSSBM to evaluate green-innovation efficiency.

Four input indicators were selected: R&D personnel, R&D capital, non-R&D personal, and non-R&D capital.

R&D personnel: Because it is difficult to separate green innovation from traditional innovation, there are no special statistical data on green-innovation personnel and funds. The full-time equivalent of R&D personnel was used.

R&D capital: The intramural expenditure on R&D were selected and the method of Li et al. [53] to calculate the stock of the internal expenditure of R&D funds was used with a base period of 2006.

Non-R&D personnel: Non-R&D personnel mainly focuses on the learning-by-doing effect from the accumulated experience of production personnel. Therefore, the average number of workers employed in above-scale enterprises is represented as the non-R&D personnel.

Non-R&D capital: Non-R&D capital includes expenditure on introducing technology, on digestion and absorption, on purchasing domestic technology, and on technological transformation. Meanwhile, the GDP price index was used to measure its stock estimation.

The outputs included expected outputs and non-expected outputs:

Expected outputs: The two expected outputs were the number of granted patents and the sales revenue of new products. Among them, the sales revenue of new products was deflated using the producer price index.

Non-expected outputs: The non-expected outputs represent negative environmental externalities in the process of green innovation. Scholars generally select the emissions of three industrial wastes as pollution indicators. Because industrial solid waste in China has been mostly disposed of and utilized in recent years, the amount of dumped and discarded industrial solids in 2016 only accounted for 0.01%. This might not reflect the actual environmental situation if solid-waste emissions are used as a non-expected output. Based on the data availability and the treatment of pollutants, industrial $SO_2$ emissions and industrial-wastewater COD emissions were selected as the non-expected outputs.

### 4.1.3. Explanatory Variable—Internet

Based on the literature review, we found that the single indicators mainly refer to the utilization level and infrastructure of the internet. The single indicators reflecting the utilization level of internet are commonly the internet utilization rate and the number of internet users; the single indicators reflecting the infrastructure of internet are commonly the length of optical-cable internet lines and the number of internet-access ports [4,23]. At present, it is generally believed that the internet is increasingly integrated with all aspects of economy, society, and life, which has a huge effect on real economy. Compared with the utilization level, the infrastructure level of the internet is more significant. In particular, China's 14th Five-Year Plan clearly proposes the systematic layout of new infrastructure and accelerates the construction of fifth-generation mobile communications, industrial internet, and big-data centers. Moreover, the internet-utilization level depends on the internet-infrastructure development. Therefore, we selected the internet infrastructure as the evaluation indicator for the internet. Based on the data availability, we used the number of internet-access ports to measure the internet according to the research of Cui et al. [4].

### 4.1.4. Control Variables

Government scale: The size of local government can affect the industrial structure by increasing industrial demand. Thus, this paper used the ratio of government public-budget revenue to GDP to express government scale.

Marketization degree: Based on the principle of data availability and regional comparability, the proportion of employment in non-state-owned enterprises in total employment was used to measure the level of marketization.

Opening up: The ratio of import and export volume to GDP can be measured to evaluate the effect of the foreign funds on local industries.

Human capital: It is generally believed that the more years of education, the higher the level of human capital. Thus, we used the average years of education in each province to express the level of human capital.

Regional economy: There is no doubt that the level of economic development is the core factor influencing industrial upgrading. Therefore, regional economy was measured by per capita GDP in this paper.

Urbanization level: The increased level of urbanization can enable the reallocation of resources, which is one of the most important factors influencing the change of industrial structure. Thus, the proportion of urban population in total population was expressed as the urbanization level.

### 4.1.5. Data Sources

Due to the availability of data, this paper selected panel data from 30 provinces and cities in China that do not include the autonomous regions of Tibet and Hong Kong, Macao, and Taiwan, from 2006 to 2019. In addition, the data were primarily drawn from the China Statistical Yearbook, China Science and Technology Statistical Yearbook, China Energy Yearbook, and China Environment Statistical Yearbook. For missing data, we adopted the trend-interpolation method for processing. Due to the large values of the internet, this paper adopted a logarithmic form for this indicator to reduce the absolute difference between the data and to eliminate heteroscedasticity. The absolute differences of the human-capital indicator and regional-economy indicator were relatively small, and other indicators were in the form of ratios, so we did not need to be logarithmic.

The indicators used in this study are described in Table 1.

**Table 1.** The descriptive statistics of the indicators.

| Variable | Definition | Obs | Mean | Std.Dev. | Min | Max |
|---|---|---|---|---|---|---|
| upgrade | Industrial upgrading | 420 | 0.478 | 0.127 | 0.000 | 0.856 |
| ssbm | Green innovation | 420 | 0.457 | 0.269 | 0.067 | 1.125 |
| dig | Internet | 420 | 0.661 | 0.123 | 0.273 | 0.905 |
| gov | Government scale | 420 | 0.242 | 0.109 | 0.095 | 0.758 |
| mar | Marketization degree | 420 | 0.522 | 0.160 | 0.236 | 0.868 |
| open | Opening up | 420 | 0.309 | 0.355 | 0.013 | 1.711 |
| edu | Human capital | 420 | 8.917 | 0.979 | 6.594 | 12.681 |
| pgdp | Regional economy | 420 | 3.672 | 2.605 | 0.610 | 15.476 |
| urb | Urbanization level | 420 | 0.547 | 0.137 | 0.275 | 0.942 |

*4.2. Spatial Econometrics Method*

4.2.1. Setting of the Spatial Weight Matrix

The selection of spatial weight matrices can affect the final analysis results [51]. Thus, in view of the sensitivity of the spatial econometric model to the spatial weight matrix, we adopted the four most common weight matrices that are used by researchers at present to estimate the effect of the internet and green innovation on industrial upgrading. The first is the adjacency matrix ($W_1$), the second is the geographical matrix ($W_2$), and the third is the economic matrix ($W_3$) in terms of economic significance. Meanwhile, considering the limitation of the weight matrix constructed simply by geographical distance or economic distance, the nested matrix of geographical and economic distance ($W_4$) was also constructed.

$$W_1 = \begin{cases} 1, & \text{when region } i \text{ is adjacent to } j \\ 0, & \text{when region } i \text{ is not adjacent to } j \end{cases} \tag{6}$$

$$W_2 = \begin{cases} \frac{1}{d_{ij}^2}, & i \neq j \\ 0, & i = j \end{cases} \tag{7}$$

$$W_3 = \begin{cases} \frac{1}{|\overline{Y}_i - \overline{Y}_j|}, & i \neq j \\ 0, & i = j \end{cases} \tag{8}$$

$$W_4 = W_2 \cdot diag(\frac{\overline{Y}_1}{\overline{Y}}, \frac{\overline{Y}_2}{\overline{Y}}, \cdots, \frac{\overline{Y}_n}{\overline{Y}}) \tag{9}$$

$$d_{ij} = ar\cos[(\sin\phi_i \times \sin\phi_j) + (\cos\phi_i \times \cos\phi_j \times \cos\Delta\tau)] \times R \tag{10}$$

where $d_{ij}$ is the geographical distance between region $i$ and region $j$, $\phi_i$ ($\phi_j$) denotes the latitude and longitude of region $i(j)$, $\Delta\tau$ is the difference in longitude between two regions, $R$ is the radius of the ball, $\overline{Y}_i$ is the annual average value of per capita GDP in region $i$ from 2006 to 2019 (data source: China Statistical Yearbook), and $n$ is the number of regions.

4.2.2. Spatial Autocorrelation Test

Before spatial econometric measurement, the spatial autocorrelation test among variables needed to be carried out. At present, Moran's I is commonly adopted to verify the spatial autocorrelation of variables, and the calculation formulas are as follows:

$$I = \frac{n\sum_{i=1}^{n}\sum_{j=1}^{n}W_{ij}(x_i - \overline{x})(x_j - \overline{x})}{\sum_{i=1}^{n}\sum_{j=1}^{n}W_{ij}\sum_{i=1}^{n}(x_i - \overline{x})^2} \tag{11}$$

$$I_i = \frac{n(x_i - \overline{x})\sum_j W_{ij}(x_j - \overline{x})}{\sum_i (x_i - \overline{x})^2} \tag{12}$$

where $I$ is the value of the global Moran's I index, $I_i$ is the local Moran's I for region $i$, $x_i$ is the value of a variable in region $i$, $\bar{x}$ is the average value of all samples, and $w_{ij}$ is a spatial weight matrix between region $i$ and $j$.

### 4.2.3. Spatial Panel Measurement

The spatial models were applied after the spatial autocorrelation test. The main types of spatial econometric models have a spatial autoregressive model (SAR), spatial error model (SEM), and spatial Durbin model (SDM). Because the test result of model selection in Section 5.2. was SEM, we only introduced SEM, as stated in the following equation:

$$upgrade_{it} = c + \beta X_{it} + \varepsilon_{it} \tag{13}$$

where $upgrade_{it}$ is the industrial upgrading of the region $i$ in the $t - th$ year, $X_{it}$ represents the core explanatory variables and control variables, and $\varepsilon_{it}$ is a random disturbance term.

## 5. Empirical Results and Analysis

### 5.1. Spatial Autocorrelation Analysis

Because the nested matrix included geographical distance and economic distance, we only showed the global Moran's I value based on two spatial weight matrixes (Table 2). As shown in Table 2, all values of core variables were greater than 0 and most *p*-values were less than 0.1 except for a few years. Compared with the adjacency matrix, all *p*-values were less than 0.1 with the nested matrix. This indicates that the internet, green innovation, and industrial upgrading had significant positive spatial correlation at the 0.01 level of significance during 2006–2019.

**Table 2.** The global Moran's I of core variables.

| | Adjacency Matrix | | | Nested Matrix | | |
| --- | --- | --- | --- | --- | --- | --- |
| | Upgrade | ssbm | dig | Upgrade | ssbm | dig |
| 2006 | 0.115 | 0.322 *** | 0.209 ** | 0.124 ** | 0.109 * | 0.193 *** |
| 2007 | 0.189 ** | 0.354 *** | 0.185 ** | 0.183 *** | 0.128 ** | 0.188 *** |
| 2008 | 0.143 * | 0.311 *** | 0.165 ** | 0.109 ** | 0.100 ** | 0.178 *** |
| 2009 | 0.209 ** | 0.439 *** | 0.214 ** | 0.179 *** | 0.246 *** | 0.191 *** |
| 2010 | 0.257 *** | 0.432 *** | 0.190 ** | 0.201 *** | 0.256 *** | 0.184 *** |
| 2011 | 0.240 ** | 0.422 *** | 0.154 * | 0.190 *** | 0.203 *** | 0.154 ** |
| 2012 | 0.273 *** | 0.486 *** | 0.148 * | 0.214 *** | 0.262 *** | 0.167 ** |
| 2013 | 0.226 ** | 0.456 *** | 0.102 | 0.170 ** | 0.265 *** | 0.127 ** |
| 2014 | 0.253 *** | 0.521 *** | 0.094 | 0.180 *** | 0.306 *** | 0.128 ** |
| 2015 | 0.220 ** | 0.420 *** | 0.126 * | 0.164 ** | 0.245 *** | 0.121 ** |
| 2016 | 0.214 ** | 0.399 *** | 0.093 | 0.153 ** | 0.221 *** | 0.113 * |
| 2017 | 0.219 ** | 0.482 *** | 0.101 | 0.155 ** | 0.274 *** | 0.118 ** |
| 2018 | 0.228 ** | 0.281 *** | 0.102 | 0.160 ** | 0.143 ** | 0.119 *** |
| 2019 | 0.223 ** | 0.356 *** | 0.098 | 0.151 ** | 0.212 *** | 0.112 * |

***, **, and * indicate significance at the 1%, 5%, and 10% levels, respectively.

Furthermore, we used STATA software to calculate the local Moran's I of the core variables based on the nested matrix (shown in Figures 2–4). It should be noted that the numbers 1 to 30 in Figures 2–4 represent Beijing, Tianjin, Hebei, Shanghai, Inner Mongolia, Liaoning, Jilin, Heilongjiang, Shanxi, Zhejiang, Jiangsu, Anhui, Fujian, Jiangxi, Shandong, Henan, Hubei, Hunan, Guangdong, Guangxi, Hainan, Chongqing, Sichuan, Guizhou, Yunnan, Shaanxi, Gansu, Qinghai, Ningxia, and Xinjiang, respectively.

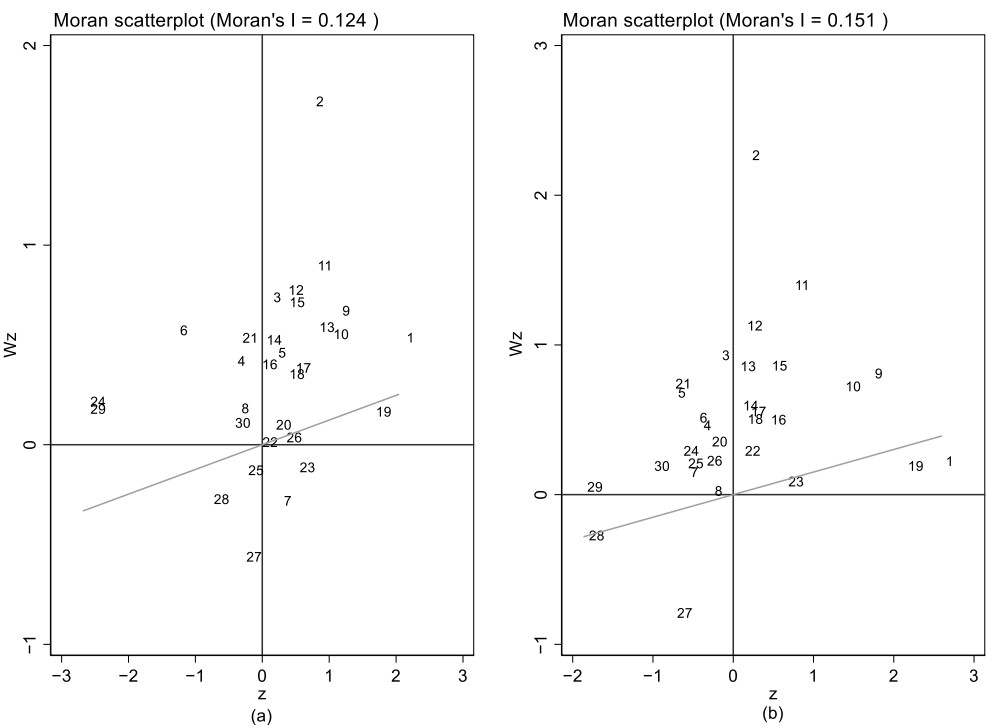

**Figure 2.** Scatter plot of local Moran's I of industrial upgrading in 2006 (**a**) and 2019 (**b**).

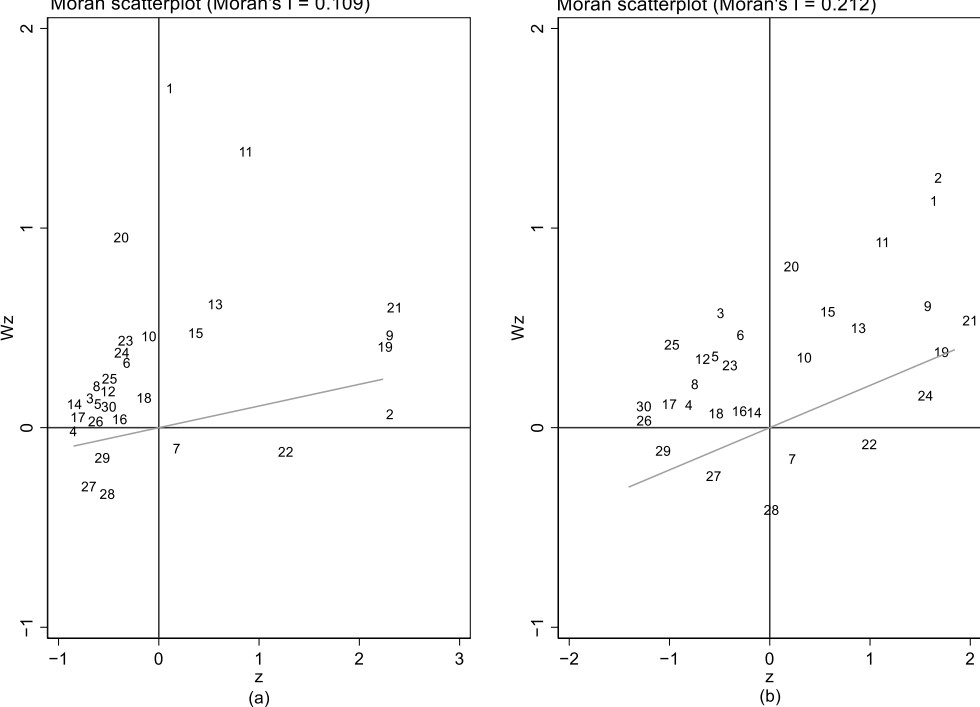

**Figure 3.** Scatter plot of local Moran's I of green innovation in 2006 (**a**) and 2019 (**b**).

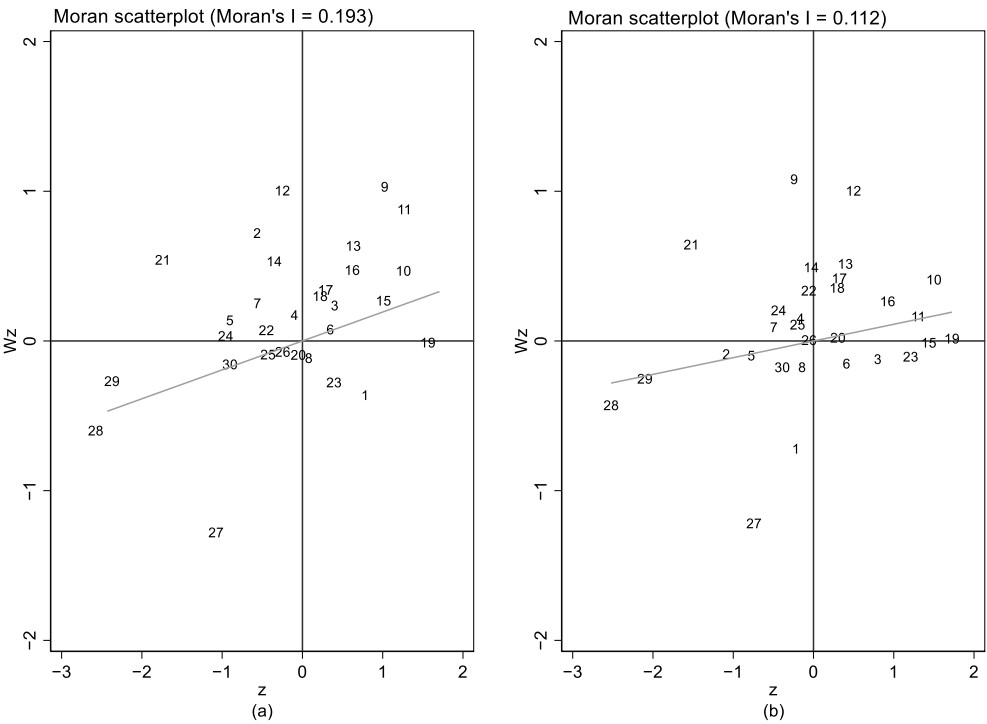

**Figure 4.** Scatter plot of local Moran's I of the internet in 2006 (**a**) and 2019 (**b**).

For industrial upgrading, most of the regions represented high–high agglomeration trends, followed by low–high agglomeration. These regions were mainly located in the central and eastern regions of China. According to the spatial distribution, we can conclude that the advanced provinces in the eastern area might have had a siphoning effect on the western neighboring provinces. For green innovation, it was similar to industrial upgrading in spatial distribution. They mainly showed spatial trends of high–high and low–high agglomeration. However, the difference between the eastern region and central region is that most regions in central region represented low–high agglomeration spatial trends. From the perspective of dynamic evolution, the main change is that some low–high agglomeration areas turned into high–high agglomeration areas. To be specific, Jiangsu, Guizhou, Guangxi, and so on, with low–high agglomeration in 2006, turned into high–high agglomeration in 2019. This indicates that the gap in the green-innovation level among provinces and cities in China was reduced. The spatial agglomeration of the internet mainly represents three trends, high–high agglomeration, low–high agglomeration, and low–low agglomeration. Among them, the number of regions shown as high–high agglomeration and low–high agglomeration decreased and the number of regions shown as low–low agglomeration increased from four to eight. Considering that the regions with high–high agglomeration were mainly located in the Yangtze Delta region, the internet network construction system covering all regions can be accelerated by giving way to the leading role of the Yangtze Delta region.

### 5.2. Model Selection

Because commonly used spatial economic models include SAR, SEM, and SDM, we needed to conduct some tests to determine the specific form of the spatial measurement model. Firstly, referring to the suggestion of Anselin et al. [54], the Lagrange multiplier (LM) statistic and its robust form (robust LM) were used to determine the specific spatial measurement model. Secondly, the Hausman test was used to confirm whether to use the fixed effect or random effect for the evaluation model. The results are shown in Table 3.

**Table 3.** Spatial econometric test results.

| | Test | | Statistic | *p* | Result |
|---|---|---|---|---|---|
| | | Moran 's I | 511.827 | 0.000 | |
| LM test | Spatial error | LM | 251.820 | 0.000 | SEM |
| | | Robust LM | 241.226 | 0.000 | |
| | Spatial lag | LM | 12.266 | 0.000 | |
| | | Robust LM | 1.672 | 0.196 | |
| | Hausman test | | −542.83 | | fix effect |

The results show that LM spatial error, robust LM spatial error, and LM spatial lag were significant at the 1% level of significance, but robust LM spatial lag did not pass the significance level test of 10%. This indicates that the sample data in this paper tended to use an SEM. According to the results of the Hausman test, the statistic was negative, indicating that the fixed effect was more suitable for this study.

*5.3. Spatial Regression Results*

5.3.1. Analysis for the Whole Sample

The estimation results of the whole samples based on four spatial weight matrices are shown in Table 4 for maximum likelihood estimation method.

**Table 4.** SEM model-estimation results of the whole samples.

| Variable | Adjacency Matrix | | Geographical Matrix | | Economic Matrix | | Nested Matrix | |
|---|---|---|---|---|---|---|---|---|
| | **Model I** | **Model II** | **Model III** | **Model IV** | **Model V** | **Model VI** | **Model VII** | **Model VIII** |
| ssbm | 0.386 *** | | 0.326 ** | | 0.409 ** | | 0.350 ** | |
| dig | 0.420 *** | | 0.447 *** | | 0.586 *** | | 0.435 *** | |
| ssbm ×dig | | 0.532 *** | | 0.612 *** | | 0.641 *** | | 0.596 *** |
| gov | −0.056 | −0.258 *** | −0.039 | −0.259 *** | 0.023 | −0.268 *** | −0.050 | −0.269 *** |
| mar | 0.127 *** | 0.146 *** | 0.130 *** | 0.147 *** | 0.074 * | 0.111 ** | 0.127 *** | 0.140 *** |
| open | 0.108 *** | 0.150 *** | 0.103 *** | 0.142 *** | 0.087 *** | 0.152 *** | 0.097 *** | 0.139 *** |
| edu | 0.033 *** | 0.034 *** | 0.029 *** | 0.030 *** | 0.017 ** | 0.028 *** | 0.029 *** | 0.029 *** |
| pgdp | 0.014 *** | 0.008 *** | 0.016 *** | 0.009 *** | 0.019 *** | 0.011 *** | 0.015 *** | 0.008 *** |
| urb | −0.317 *** | −0.312 *** | −0.291 *** | −0.265 *** | −0.190 *** | −0.294 *** | −0.272 *** | −0.244 *** |
| spatial lambda | −0.316 *** | −0.396 *** | −0.261 ** | −0.435 *** | −0.456 *** | −0.137 | −0.757 *** | −0.992 *** |
| variance sigma2_e | 0.003 *** | 0.003 *** | 0.003 *** | 0.003 *** | 0.002 *** | 0.002 *** | 0.003 *** | 0.003 *** |
| R-squared | 0.873 | 0.848 | 0.874 | 0.847 | 0.873 | 0.850 | 0.873 | 0.846 |

***, **, and * indicate significance at the 1%, 5%, and 10% levels, respectively.

The results in Table 4 show that although the coefficients of the estimation results based on the four weight matrixes were different, the direction and significance did not change significantly, indicating that the research results of this paper are relatively robust and there are significant spatial dependencies.

In terms of green innovation, the estimated coefficients were all significantly positive, indicating that Hypothesis 2 is correct. In terms of the internet, the coefficients were significantly positive in models I, III, V and VII, indicating that Hypothesis 1 is correct. Compared with the coefficients of the internet and green innovation, the internet had a greater impact for industrial upgrading no matter what kind of spatial weight matrix was used. Compared with the four spatial weight matrixes, the independent effects of the internet and green innovation were highest based on the economic matrix, indicating that the spatial effects of the internet and green innovation on industrial upgrading are

closely related to the economic development. Furthermore, the estimated coefficients of the cross-term of the internet and green innovation were also significantly positive, indicating that Hypothesis 3 was proven to be correct. According to the coefficients, we can see that the coefficients of the cross-term were higher compared to the coefficients of the internet and green innovation.

In terms of control variables, except for the government-scale variable, the rest of the variables passed the significance-level test, of which the urbanization level was significantly negative. Two aspects may explain this negative effect: One is that migrant workers are mainly employed in industries with high pollution and energy consumption, which are the focus of environmental governance [55], and the other is that the uncoordinated development of land urbanization and population urbanization has caused resource misallocation.

### 5.3.2. Analysis for Regional Sample

In view of the differences in resource endowments and economic development among regions in China, this paper discusses the regional heterogeneity of the effects of the internet and green innovation on industrial upgrading across the eastern, central, and western regions. Due to space limitations, only the estimation results based on the nested matrix are reported here (Table 5).

**Table 5.** SEM model-estimation results of regional samples.

| Variable | Eastern Region | | Central Region | | Western Region | |
|---|---|---|---|---|---|---|
| | Model I | Model II | Model III | Model IV | Model V | Model VI |
| ssbm | 0.254 * | | 0.138 *** | | 0.122 *** | |
| dig | 0.110 *** | | 0.219 *** | | 0.381 *** | |
| ssbm×dig | | 0.408 ** | | 0.394 *** | | 0.375 *** |
| gov | −0.564 *** | −0.522 *** | −0.218 | −0.238 | −0.283 *** | −0.172 *** |
| mar | 0.267 *** | 0.266 *** | 0.062 | 0.081 | 0.005 | 0.012 |
| open | 0.193 *** | 0.218 *** | −0.189 ** | −0.115 | −0.039 | −0.135 * |
| edu | 0.109 *** | 0.098 *** | −0.028 *** | −0.034 *** | −0.001 | 0.004 |
| pgdp | 0.006 | 0.007 ** | 0.006 | 0.003 | 0.005 | 0.018 ** |
| urb | −0.823 *** | −0.816 *** | 0.232 *** | 0.263 ** | −0.041 | −0.102 |
| spatial lambda | −0.320 | −0.202 | −0.068 | −0.259 * | −1.037 *** | −0.947 *** |
| variance sigma2_e | 0.002 *** | 0.002 *** | 0.001 *** | 0.001 *** | 0.002 *** | 0.002 *** |
| R-squared | 0.961 | 0.956 | 0.831 | 0.681 | 0.800 | 0.841 |

***, **, and * indicate significance at the 1%, 5%, and 10% levels, respectively.

The results in Table 5 show that the evaluation results of the internet, green innovation, and their cross-term on industrial upgrading were similar in the three regions of east, central, and west. Regardless of the region, the internet, green innovation, and their cross-term can play a positive role in promoting industrial upgrading. The main reason might be that China implemented the internet and green innovation as national strategies in 2015, and local governments have responded to the national call to enhance industrial upgrading [36].

According to the coefficients of the internet, green innovation, and their cross-term in the three regions, it can be seen that the cross-term coefficients of the eastern and central regions exceeded the coefficients of the internet and green innovation, which indicates that the cooperation of the internet and green innovation was better in the eastern and central regions compared with the western region. The coefficients of the internet, green innovation, and their cross-term in the west were 0.112, 0.381, and 0.375, respectively, indicating that the interactive effect was more significant than the green-innovation independent effect but lower than the internet independent effect. The main reason might be that the development of green innovation lags behind the internet and the gap between the internet and green innovation can inhibit the multiplier effect of cooperation. In addition, due the

coefficients of the internet and green innovation in the eastern region being 0.254 and 0.110, respectively, green innovation plays a stronger role than the internet. In the eastern region, the development of the internet should be accelerated while maintaining the steady growth of green innovation. However, in the central region, the coefficient of green innovation was 0.138 less than the internet coefficient of 0.219, indicating that green innovation should be accelerated while maintaining the steady growth of the internet so as to promote the high-quality and sustainable development of industries. Combined with the above analysis, the government of the western region has to implement polices on green innovation more vigorously and urgently than the central region.

## 6. Discussion and Conclusions

### 6.1. Discussion

Previous studies have explored the relationship among the internet, green innovation, and industrial upgrading, but these studies mainly focused on the effect of the internet on industrial upgrading, the effect of green innovation on industrial upgrading, and the effect of the internet on green innovation. Thus, we incorporated the internet, green innovation, and industrial upgrading into the same research framework to analyze the theoretical mechanisms of internet, green innovation on industrial upgrading, the interaction mechanisms of internet and green innovation, and their interaction on industrial upgrading. Secondly, previous studies have not considered the impact of the spatial spillover effect of green innovation on industrial upgrading. Therefore, we constructed a spatial econometric model to study their independent and interactive effects on industrial upgrading. Meanwhile, we analyzed the regional differences across the three eastern, central, and western regions.

### 6.2. Main Conclusions

Based on the relative theory and literature analysis, we used various provincial panel data from 2006 to 2019 and four spatial weight matrices to empirically analyze the independent and interactive effects of the internet and green innovation on industrial upgrading. The main conclusions are as follows:

1.  The internet, green innovation, and industrial upgrading had significant positive spatial correlation during the study period; namely, their development was affected by both the local and adjacent areas. The main spatial agglomeration trends on the internet, green innovation, and industrial upgrading were shown as spatial trends of high–high and low–high agglomeration. Among them, most of the regions represented high–high agglomeration trends for industrial upgrading and were mainly located in the eastern and central regions. This is similar to the internet, which mainly presented high–high agglomeration trends. Most regions with high–high agglomeration were located in the Yangtze Delta region. On the other hand, green innovation mainly presented low–high agglomeration spatial trends. During the period, most regions changed from low–high agglomeration to high–high agglomeration and from low–low agglomeration to low–high agglomeration.

2.  In the independent-effect test, both the internet and green innovation significantly enhanced industrial upgrading. No matter which spatial weight matrix was used, the internet coefficient was significantly greater than the green innovation coefficient. This indicates that the internet independent effect on industrial upgrading is higher than the green-innovation independent effect. The cross-term coefficients of the internet and green innovation based on the four spatial weight matrixes were significantly positive, indicating that there is an interactive effect between the internet and green innovation on industrial upgrading. Besides, the cross-term effects were similar to the independent effects of the internet and green innovation, which are more closely related to the economic development among in the four spatial weight matrixes.

3.  Regardless of the region, the internet, green innovation, and their cross-term played a positive role in promoting industrial upgrading. However, there were still difference in the three regions when it came to the independent effects and interactive effect. Only

in the western region was the independent effect of the internet more significant than the synergistic effect. Based on the coefficients of the internet and green innovation in the western region, the development of green innovation lagged behind the internet. This was similar in the central region; the development of the internet was better than green innovation. In the eastern region, the coefficient of the internet was lower than green innovation, which was different from the central and western regions.

### 6.3. Policy Recommendations

The following policy recommendations are put forward to formulate reasonable internet policies and green-innovation measures in order to achieve industrial upgrading:

1.  Continuously and efficiently promote the construction, promotion, and application of the internet. The government should increase investment in internet-infrastructure construction, encourage and guide enterprises to "touch the internet" and "use the internet", improve the construction of supporting systems in standards and security such as data property right protection and platform architecture, and build an industrial-internet network-construction system covering all regions and industries.
2.  Optimize the green innovation environment. The government should accelerate the establishment and optimization of a cooperative-innovation mechanism of government, industry, university, research, and application; strengthen the protection of intellectual property rights; and guide enterprises to meet market demands by providing products or services to form a model of market-demand-oriented technological innovation to make up for the shortcomings of green innovation and enhance the ability of green innovation.
3.  Take the "Internet + green innovation" strategy as a foothold to fully give way to the interactive effect of internet and green innovation. The government should accelerate the formulation of "Internet + green innovation" support policies, build a multidirectional collaborative innovation platform for the internet plus green innovation, and vigorously guide social capital to invest in the core areas of internet plus green innovation spillover so as to achieve high-quality integration of the internet plus green innovation.
4.  Implement internet-plus-green-innovation policies adapted to local conditions. For the eastern region, the development of the internet should be accelerated to improve the synergistic effect between the internet and green innovation, whereas the development of green innovation should be accelerated to enhance the effect of the internet plus green innovation in the western and central regions.
5.  Pay attention to the initiative and flexibility of the internet and green innovation in industrial restructuring to maximize the synergy effect. The internet and green innovation can show a dynamic characteristic of a positive and increasingly marginal effect with the improvement of industrial development. Thus, governments at all levels should adhere to the principle of dynamic management, adjust their policy implementation efforts to maintain a dynamic balance between the internet and green innovation, and promote the maximization of synergies.

### 6.4. Limitations and Future Directions

This research was subject to some limitations, which should be considered for further research. First, when studying the impact of industrial greening, we only used total-energy-consumption reduction as a proxy indicator and did not consider other aspects such as resource utilization and pollution emissions. The next step is to establish a reasonable indicator system and select a reasonable indicator-integration method that can fully reflect the connotation of industrial greening and highlight the fact that different dimensions of industrial upgrading are complementary and equally important. Secondly, the article only studied the overall perspective of green innovation and internet on industrial upgrading. However, green innovation is a multi-stage input–output process. The independent and

interactive effects of the internet and green innovation on industrial upgrading in different green stages can be explored further.

**Author Contributions:** Data curation, Y.R.; writing, Y.R. and L.T. All authors have read and agreed to the published version of the manuscript.

**Funding:** This research received no external funding.

**Institutional Review Board Statement:** Not applicable.

**Informed Consent Statement:** Not applicable.

**Data Availability Statement:** Data can be provided by the corresponding author on request.

**Conflicts of Interest:** The authors declare no conflict of interest.

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
