# Peer review of "Internet, Green Innovation and Industrial Upgrading"

_sustainability, doi:10.3390/su142013687_

Round 1
Reviewer 1 Report
The topic of Internet, Green Innovation and Industrial Upgrading is interesting to the journal. For this reason, I recommend the publication in Sustainability Journal. Overall, the article is very interesting, and signifies an opportunity for researchers to provide a quick image of the implications of the findings of the research conducted. Hopefully, the following comments are intended to be useful:
1. Introduction
The introduction is well organize and explains the main objective of the research. In addition, it uses updated references that are of great help. This section is well structured and contains the necessary relevant information. The end of the introduction is valued positively, where the authors highlight the contributions of the paper.
2. Theoretical Mechanisms and Theoretical Hypotheses
The theoretical background is well written in general although I believe that there should be some theory to support the study. However, I suggest that the article cam improve by adding the literature review section examining Internet, Green Innovation and Industrial Upgrading. The justification of reviewing literature for examining the research questions and equations is important. The literature review should be sufficiently developed and referenced for all variables treated. However, there is a lack of an introductory paragraph that links the three subsections(2-1 to 2-3) on the basis of theoretical background and provides a research guiding line. Please add this and polish this for the article.
3. Model Establishment and Variable Measurement
The model establishment and variable measurement process carried out to produce the article is very well explained and is understandable for researchers who are both familiar with the subject matter. However, they should report on how the data matrix has been prepared and whether or not the missing data have been removed. They must also report on how the calibration of the factors formed through items has been done. Please improve this.
4. Empirical Results and Analysis
The article discusses the existence of several existing conditions in the model. The article also identify, explain and discuss its significance implications for the study. Well done.
5. Discussion and Suggestions
The conclusions suggestions are well summarized. However, the discussion should be improved to compare the results with other studies. In its current form it does not discuss the results. I would recommend indicating where this research should be heading in the future in which direction. By analyzing the implications of this study the authors should add one or two paragraphs at the end of the conclusions regarding the implications.
Author Response
Dear Reviewers of Sustainability,
We the author of the manuscript titled " Internet,Green Innovation and Industrial Upgrading" would like to express our appreciation and thank you for your time in reviewing our manuscript. We have incorporated most of the suggestions made by you and other reviews. Any revisions to the manuscript have been marked up using the “Track Changes” function.
We provide a cover letter to explain, point by point, the details of the revisions to the manuscript and your responses to the referees’ comments. Please see the attachment!
Best regards.
Yours sincerely.
Rong Yafei. Tuesday, October 18, 2022.

Reviewer 2 Report
This manuscript deals with a relevant issue: the relations between Internet,Green Innovation and Industrial Upgrading.
The authors analyze the spatial correlation between Internet,Green Innovation and Industrial Upgrading and agglomeration characteristics on 30 provinces in China from 2006 to 2019.
The research issue and objectives are clear and coherently stated.
The research method and procedures are robust and clearly stated.
The research results and discussion are soundly presented and diuscussed.
The conclusions and policy implications are pertinent and supported by the research results.
Two minor points are worth revisiting:
1 - In the Abstract, it seems relevant to add the research methodological contributions and the main policly implications;
2 - From line 316 to line 316 it stated that "Because consumption reduction is the top priority compared with other aspect of industrial greening, in addition, consumption reduction can affect the level of environmental pollution through the needed clean technologies [64]." The use of "consumption reduction" as a proxi for "industrial greening" is worth justifying/discussing a little more.
Author Response

(The authors gave the same response as above.)

Reviewer 3 Report
The author evaluates the effect of internet and green innovation on industrial upgrading in China based on historical data, and provides interesting discussion. Meanwhile, the manuscript needs modification before publication. The detailed comments are as followed:
1. The concept and scope of internet, green innovation and industrial upgrading should be defined and explained at the first place.
2. The literature review is not enough as it does not provide convincing evidence about the interaction between internet and green innovation. I think theoretical explanation and real examples are both necessary.
3. Figure 1 is not logically convincing and some of the concepts are terribly translated. For example, what is market need effect? It is very hard to understand. Besides, errors exist like "need feeedback".
4. In Equation (10), the variables IS, II, IC, IG should be explained. What are their definitions? What statistical data are used to represent these variables?
5. Comparison between the models listed in Table 4 should be discussed.
Author Response

(The authors gave the same response as above.)

Reviewer 4 Report
Congratulations to the authors!
In my opinion, the paper is suitable for publication in the actual form.
Author Response
Dear Reviewer of Sustainability,
We the author of the manuscript titled " Internet,Green Innovation and Industrial Upgrading" would like to express our appreciation and thank you for your time in reviewing our manuscript.
Thank you for giving us the opportunity to submit a revised draft of our manuscript for publication in the Journal of Sustainability. Thank you once again as we hope for your time again to check the revised version of our manuscript.
Best regards.
Yours sincerely.
Rong Yafei. Monday, October 17, 2022.
Round 2
Reviewer 3 Report
The author has improve the manuscript according to my previous comments. It can be published after careful checking of language.